

# A practical guide and power analysis for GLMMs: detecting among treatment variation in random effects

Morgan P. Kain[1], Ben M. Bolker[2] and Michael W. McCoy[1]

[1] Department of Biology, East Carolina University, Greenville, NC, USA
[2] Departments of Mathematics & Statistics and Biology, McMaster University, Hamilton, ON, Canada

## ABSTRACT

In ecology and evolution generalized linear mixed models (GLMMs) are becoming increasingly used to test for differences in variation by treatment at multiple hierarchical levels. Yet, the specific sampling schemes that optimize the power of an experiment to detect differences in random effects by treatment/group remain unknown. In this paper we develop a blueprint for conducting power analyses for GLMMs focusing on detecting differences in variance by treatment. We present parameterization and power analyses for random-intercepts and random-slopes GLMMs because of their generality as focal parameters for most applications and because of their immediate applicability to emerging questions in the field of behavioral ecology. We focus on the extreme case of hierarchically structured binomial data, though the framework presented here generalizes easily to any error distribution model. First, we determine the optimal ratio of individuals to repeated measures within individuals that maximizes power to detect differences by treatment in among-individual variation in intercept, among-individual variation in slope, and within-individual variation in intercept. Second, we explore how power to detect differences in target variance parameters is affected by total variation. Our results indicate heterogeneity in power across ratios of individuals to repeated measures with an optimal ratio determined by both the target variance parameter and total sample size. Additionally, power to detect each variance parameter was low overall (in most cases >1,000 total observations per treatment needed to achieve 80% power) and decreased with increasing variance in non-target random effects. With growing interest in variance as the parameter of inquiry, these power analyses provide a crucial component for designing experiments focused on detecting differences in variance. We hope to inspire novel experimental designs in ecology and evolution investigating the causes and implications of individual-level phenotypic variance, such as the adaptive significance of within-individual variation.

Corresponding author
Morgan P. Kain,
morganpkain@gmail.com

## INTRODUCTION

Recent advances in computing power and access to increasingly sophisticated statistical tools such as generalized linear mixed effects models are changing research in ecology, evolution and behavior. Research questions and data analyses are no longer confined to the assumptions of clean experimental designs based on agricultural plots and Normal error distributions. Researchers now commonly incorporate multiple levels of hierarchical nesting (e.g., repeated measures) and can analyze data using a wide array of non-Gaussian error distribution models. This change is epitomized by the recent increase in use of linear and generalized linear mixed models ([G]LMMs: *Bolker et al., 2009*; J Touchon & WM McCoy, 2014, unpublished data). These powerful tools permit appropriate modeling of variation among groups and across space and time, allowing for more accurate extrapolation of statistical results to unobserved data, as well as statistical tests of variance components (*Gelman & Hill, 2006*; *Bolker et al., 2009*; *Zuur et al., 2009*; *Zuur, Hilbe & Leno, 2013*).

The upsurge in the use of LMM and GLMM has been facilitated by several recent methods papers (*Bolker et al., 2009*; *Martin et al., 2011*; *Dingemanse & Dochtermann, 2013*; *Schielzeth & Nakagawa, 2013*) and textbooks (*Gelman & Hill, 2006*; *Zuur et al., 2009*; *Zuur, Hilbe & Leno, 2013*; *Bolker, 2015*) specifically aimed at non-statisticians. While these resources have accelerated the adoption of these tools, there are still too few resources guiding researchers through the choices that must be made *prior to* the initiation of a new experiment, such as the sampling scheme that will optimize the power of an experiment requiring analysis by linear (*Moineddin, Matheson & Glazier, 2007*; *Scherbaum & Ferreter, 2009*; *Martin et al., 2011*) and generalized linear (*Johnson et al., 2014*) mixed models. In this paper, we develop a blueprint for conducting power analyses for GLMMs using the `lme4` package (*Bates et al., 2014*) in the R statistical programming environment (*R Development Core Team, 2014*). We focus on a specific application aimed at detecting differences in variance among- and within-groups between clusters of groups, such as differences in the amount of variation among individuals (group) between the treatments (cluster) of a manipulative or observational experiment.

Power analysis is fundamental to good experimental design, but is often overlooked (*Jennions & Møller, 2003*), or in the case of GLMMs, simply too difficult to implement for many practitioners. Power analyses can be especially daunting for GLMMs because they require large simulations with complex, non-Normal and non-independent data structures (*Johnson et al., 2014*). In this paper we take advantage of recent developments in the `lme4` package in R that simplify the process of simulating appropriate data (>version 1.1–6). Despite the increasing use of GLMMs in ecology and evolution and growing interest in variance, we are aware of no papers that present power analyses for statistical tests on variance using GLMMs, and only one paper presenting power analyses for fixed effects in GLMMs (*Johnson et al., 2014*). Indeed, *Johnson et al.*'s (*2014*) analysis illustrates that power analyses conducted for hierarchically structured experiments that do not incorporate random effects can generate biased estimates of fixed effects, highlighting the need for a better understanding of these approaches.

While most applications of GLMMs to date have focused on detecting differences in fixed effects while appropriately accounting for random effects (e.g., *Johnson et al., 2014*), GLMMs are under rapid development and many new applications are now possible (e.g., modeling heterogeneous error variance: *Kizilkaya & Tempelman, 2005*; *Cernicchiaro et al., 2013*). With growing interest in variance as the parameter of inquiry (*Moore, Brodie & Wolf, 1997*; *Lynch & Walsh, 1998*; *Benedetti-Cecchi, 2003*; *Hill & Zhang, 2004*; *Nussey, Wilson & Brommer, 2007*; *Dingemanse et al., 2010*; *Tonsor, Elnaccash & Scheiner, 2013*; *Westneat, Wright & Dingemanse, 2014*), there is an increased need for accessible, flexible simulation-based power analyses that assess power to detect differences in random effects—the magnitude of variation present among repeated measures at a specific hierarchical level (*Gelman & Hill, 2006*; *Zuur et al., 2009*)—by treatment.

Here we present parameterization and power analyses for random-intercepts and random-slopes GLMMs that test for differences in variation among- and within-groups (e.g., differences in the amount of variation among- and within-individuals in different treatments of an experiment). We focus on three key parameters: (1) Among-group variation in intercept; (2) Within-group variation in intercept; (3) Among-group variation in slope. We examine each of these comparisons in two contexts. First, we describe the optimal ratio of groups (e.g., hospitals, schools or individuals) to observations within groups (e.g., patients, students, repeated observations of each individual) that maximizes power to detect differences in each variance parameter. In experiments with binomially distributed response variables, observations within groups are organized into $j$ sampling occasions, each containing $n$ Bernoulli observations (e.g., individuals are each measured $n$ times for the presence or absence of a behavior in each sampling occasion $j$). Here we discuss the ratio of groups to total observations within groups ($n * j$), and consider how varying $n$ and $j$ affect power to detect each variance parameter. Second, we explore how power to detect differences in specific variance parameters is affected by increasing variation in non-target parameters (e.g., how power to detect differences in among-group variation decreases as within-group variance increases). We consider both random-intercepts and random-slopes models because of their generality as focal parameters for most applications, and choose to focus on the extreme case of hierarchically structured binomial data because binary response data (e.g., the presence or absence of a behavior) contains the least possible amount of information per observation and yet is a common data format for a variety of endpoints measured in ecology.

We use vocabulary and examples from behavioral ecology to illustrate our models because of their immediate applicability to emerging questions in this field. Specifically, we evaluate power to detect significant differences in among-individual variation in reaction norm intercept and slope, and within-individual variation in intercept between individuals (i.e., among individuals aggregated by treatment) (*Nussey, Wilson & Brommer, 2007*; *Dingemanse et al., 2010*). Our methods extend current approaches used in behavioral ecology for quantifying among-individual variation away from simply testing whether there is significant deviation from a null model of no variation (*Martin et al., 2011*; *Van de Pol, 2012*; *Dingemanse & Dochtermann, 2013*)toward quantifying and contrasting

the magnitude of among- and within-individual variation among multiple groups of individuals.

While we focus on behavioral ecology as the primary application for these power analyses, these analyses are generally appropriate for comparing variation in hierarchically structured data. For example, similar methods could be used to evaluate power to detect the effects of a new experimental district-wide policy on variation among schools in student performance, or to evaluate variation among individuals in foraging success between populations (e.g., birds in an urban environment experience canalized behavior relative to birds in a natural environment, possibly reducing diversification; see *De León et al., 2011*).

In an effort to present a framework that is customizable for a diversity of research problems, we focus on a general sampling scheme in which several Bernoulli observations ($n > 1$) within multiple sampling occasions ($j > 1$) are available for each individual. Under this sampling scheme multiple probabilities of "success" (e.g., the probability of displaying a behavior) are available for each individual, which is necessary for quantifying within-individual variation (variation in the probability an individual displays a behavior between sampling occasions). However, we note that often in behavioral ecology only a single Bernoulli observation ($n = 1$) is available for each sampling occasion $j$. We include a description on how to modify this general case to accommodate single observations per sampling occasion in Supplemental Information 1. Finally, while we focus on the binomial GLMM, the framework presented here generalizes easily to other error distribution models such as Normal, log-Normal, or Gamma (for continuous responses) or Poisson or negative binomial (for count responses).

## METHODS

### Linear mixed model

We begin by introducing a general linear mixed model (LMM) to illustrate the variance components we are interested in (Fig. 1) and their applications in behavioral ecology. We provide only a brief introduction to LMMs here because they have been extensively discussed in several recent reviews and textbooks (*Gelman & Hill, 2006*; *Zuur et al., 2009*; *Stroup, 2012*; *Zuur, Hilbe & Leno, 2013*; *Dingemanse & Dochtermann, 2013*; *Bates et al., 2014*; *Bolker, 2015*). We use the notation of *Stroup (2012)* to facilitate a transition to the binomial GLMM model, which is the focus of our power analyses.

A two treatment linear mixed model can be written as:

$$y_{ijk}|b_{0ik}, b_{1ik} \sim \text{Normal}(\mu_{ijk}, \sigma^2_{\varepsilon k}) \tag{1}$$

$$\eta_{ijk} = \beta_{0k} + b_{0ik} + (\beta_{1k} + b_{ik})X_{ijk} \tag{2}$$

Identity link: $\eta_{ijk} = \mu_{ijk}$ (3)

$$\begin{bmatrix} b_{0ik} \\ b_{1ik} \end{bmatrix} \sim \text{MVN}\left( \begin{bmatrix} 0 \\ 0 \end{bmatrix}, \begin{bmatrix} \sigma^2_{0k} & \sigma_{01k} \\ \sigma_{0lk} & \sigma^2_{1k} \end{bmatrix} \right). \tag{4}$$

Here, a single phenotypic measurement $y_{ijk}$ is of individual $i$, at level $j$ of the covariate $X$ (in studies of animal behavior the covariate of interest is often an environmental gradient)

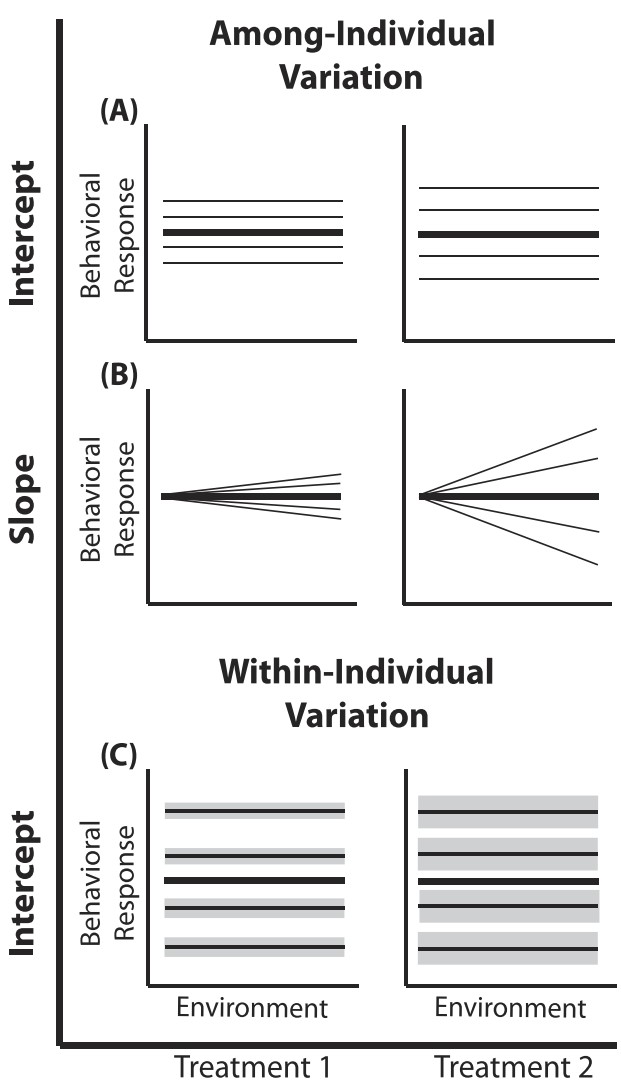

**Figure 1 Reaction norm plots for a two treatment LMM.** In all graphs bolded black lines depict treatment mean reaction norms and thin lines depict reaction norms of individuals. Grey envelopes in (C) illustrate the magnitude of within-individual intercept variation. Here among-individual variation in intercept (A), slope (B), and within-individual variation in intercept (C) is larger in treatment 2.

in treatment $k$. This model is composed of three components: the treatment mean in environment $j$ ($\beta_{0k} + \beta_{1k}X_{ijk}$), the unique average response of individual $i$ across the environmental gradient ($b_{0k} + b_{1k}X_{ijk}$), and a residual error due to the variation around the mean of individual $i(\sigma_{\varepsilon k}^2)$, which is assumed to be homogenous across $X$ and among all individuals in treatment $k$, but is allowed to vary by treatment. Individuals vary from the treatment mean reaction norm in both their intercept ($b_{0ik}$) and slope ($b_{1ik}$), which together compose the total phenotypic variance attributable to among-individual variation. This individual contribution is quantified using a random intercepts and slopes model with a multivariate Normal (MVN) distribution (4). Variation among individuals in intercept and slope are $\sigma_{0k}^2$ and $\sigma_{1k}^2$ respectively; covariance between intercept and slope is given by $\sigma_{01k}$. In a LMM, the linear predictor directly predicts the mean, as shown by the

identity link function in Eq. (3). In a GLMM, the linear predictor predicts a function of the mean $g(x)$, which must be linearized through the use of non-identity link functions; for example, we use the standard logit (log-odds) link for Binomial GLMM.

### Among-individual variation in intercept

In behavioral ecology among-individual variation in intercept $\sigma_{0k}^2$ describes the amount of variation around average behavior that occurs among individuals (Fig. 1). In field studies, $\sigma_{0k}^2$ often describes variation among individuals in their average behavior in the average environment (see *Nussey, Wilson & Brommer, 2007*; *Westneat et al., 2011*). Previous work has demonstrated that individuals from a diversity of taxa vary in their average behavior in many different environments (*Bell, Hankison & Laskowski, 2009*). Yet, comparisons of among- and within-individual variation in average behavior (or other forms of plasticity) among populations or treatments remain underrepresented (e.g., *Westneat et al., 2011*; *Dingemanse et al., 2012*). For example, *Westneat et al. (2011)* found that female house sparrows vary less from one another in their average provisioning behavior than male sparrows. In the model presented here, the random intercept ($b_{0ik}$) for each individual (e.g., male and female nest provisioning rates are drawn from Normal distributions with different variances) is drawn from a treatment-specific Normal distribution.

### Within-individual variation in intercept

Within-individual variation in intercept ($\sigma_{\varepsilon k}^2$) is defined as the amount individuals vary around their own average behavior. Within-individual variation is routinely used for the calculation of repeatability in studies of animal personality (*Bell, Hankison & Laskowski, 2009*; *Dingemanse et al., 2010*) or more often is simply regarded as noise, despite the well established ecological and evolutionary implications of within-individual variation (*Stamps, Briffa & Biro, 2012*; *Biro & Adriaenssens, 2013*; *Westneat, Wright & Dingemanse, 2014*; *Cleasby, Nakagawa & Schielzeth, 2015*). For example, a variable predator environment may select for individual prey that vary greatly around their mean behavior to remain unpredictable (*Stamps, Briffa & Biro, 2012*). LMMs can directly quantify patterns of within-individual variation when repeated measures within multiple individuals are available, facilitating comparisons of consistency responses between individuals (*Dingemanse & Dochtermann, 2013*). Here we are interested in determining if $\sigma_{\varepsilon k}^2$ differs by treatment. In other words, do individuals in one population or treatment exhibit more intra-individual behavioral variation than individuals from a second population or treatment?

### Among-individual variation in slope

Substantial empirical work has shown that individual animals in a variety of taxa display variation in phenotypic plasticity (*Martin & Réale, 2008*; *Mathot et al., 2011*; *Dingemanse et al., 2012*); using mixed models to quantify this variation has been the primary focus of several recent papers (*Martin et al., 2011*; *Van de Pol, 2012*; *Dingemanse & Dochtermann, 2013*). Among-individual variation in phenotypic plasticity has implications for the rate of evolutionary change, population stability and population persistence (*Wolf & Weissing, 2012*; *Dingemanse & Wolf, 2013*); thus defining those populations exhibiting greater

individual variation in plasticity could help distinguish stable populations and populations with a high probability of micro-evolutionary change (*Pigliucci, 2001*; *Ghalambor, Angeloni & Carroll, 2010*). To quantify group differences in plasticity variation, multiple measurements within each individual across an environmental gradient are required. Here we are interested in determining if $\sigma_{1k}^2$ differs by treatment.

## Binomial GLMM

We assess power of a binomial GLMM for detecting differences in variation by treatment. This model can be written as:

$$y_{ijk}|b_{0ik}, b_{1ik}, v_{ijk} \sim \text{Binomial}(N_{ijk}, \pi_{ijk}) \tag{5}$$

$$\eta_{ijk} = \beta_0 + b_{0ik} + (\beta_1 + b_{1ik})X_{ijk} + v_{ijk} \tag{6}$$

$$\text{Inverse-logit: } \pi_{ijk} = 1/(1 + e^{-\eta}ijk) \tag{7}$$

$$\begin{bmatrix} b_{0ik} \\ b_{1ik} \end{bmatrix} \sim \text{MVN} \left( \begin{bmatrix} 0 \\ 0 \end{bmatrix}, \begin{bmatrix} \sigma_{0k}^2 & \sigma_{01k} \\ \sigma_{01k} & \sigma_{1k}^2 \end{bmatrix} \right) \tag{8}$$

$$v_{ijk} \sim \text{Normal}(0, \sigma_{vk}^2). \tag{9}$$

Here, $y_{ijk}$ is the number of "successes" in $N_{ijk}$ observations of the $i$th individual in treatment $k$ at the $j$th sampling occasion. When an environmental covariate ($X$) is present, we assume one sampling occasion occurs at each level of the covariate $j$. Here, in the absence of an environmental covariate, the linear predictor reduces to $\eta_{ijk} = \beta_0 + b_{0ik} + v_{ijk}$ and the $j$th occasion is simply a repeated sampling occasion in the same conditions. Note, when $N_{ijk} = 1$ there is only 1 observation per sampling occasion $j$, making $y_{ijk}$ a Bernoulli response variable (see Supplemental Information 1). When $y_{ijk}$ is Bernoulli, overdispersion ($v_{ijk}$) and thus within-individual variation is not identifiable.

In this model $\pi_{ijk}$ describes the underlying probability of individual $i$ in treatment $k$ at occasion $j$ exhibiting a behavior. Variation in $\pi$ isdetermined by the linear combination of predictors on the logit (log-odds) scale: group intercept ($\beta_0$), group slope ($\beta_1$), individual unique intercept ($b_{0ik}$), slope ($b_{1ik}$), and observation level overdispersion that decrease predictive power at each observation ($v_{ijk}$). This linear predictor is transformed with the inverse-logit link to produce $\pi_{ijk}$, which follows a logit-Normal-binomial mixed distribution.

We use an observation-level random effect to model additive overdispersion (*Browne et al., 2005*), which models increased variance (following a Normal distribution with variance $\sigma_{vk}^2$) in the linear predictor on the link scale (*Nakagawa & Schielzeth, 2010*). Overdispersion is used to quantify within-individual variation because it models variation in $\pi$ between each sampling occasion $j$ for each individual. Here the magnitude of overdispersion is allowed to vary by treatment (for an example of multiple data sets where this occurs see *Hinde & Demétrio, 1998*), which is a focus of our power analysis.

The transformation through the inverse-logit function makes each of the three target variance components difficult to visualize with a concise figure. However, because the binomial GLMM model follows similar patterns as the LMM, we present power analyses

for the binomial GLMM using the visual aid presented for the LMM (Fig. 1). Finally, we simulate data for a fully balanced design without losing generality. See *Martin et al. (2011)* and *Van de Pol (2012)* for a discussion on experimental designs where individuals are assayed in partially overlapping environments and when only single measurements are obtained for some individuals.

## Simulations

All data were simulated in the R statistical programming environment using newly developed simulation capabilities of the `lme4` package (>version 1.1–6, *Bates et al., 2014*). Guidelines for parameterizing the GLMMs and running data simulations and power analyses are provided in Supplemental Information 1. For a given total sample size, we present simulations for determining the optimal ratio of total number of individuals versus the number of repeated measures within individuals needed to provide power to detect a difference among treatments 80% of the time. We conducted simulations for multiple ratios of individuals to total observations within individuals, varying both sampling occasions ($j$) and Bernoulli observations within sampling occasions ($n$). Next, we describe simulations that evaluate how increasing "noise" (variation in non-target random effects) affects power to detect differences in targeted variance comparisons.

For both scenarios we simulate data with biologically relevant parameter values that illustrate common trends in power. At extreme parameter values the trends presented here may not hold due to interactions between the variance components that arise at the boundaries of binomial space. We do not dwell on these exceptions since they are unrealistic for most empirical data sets, but suggest exploration of these exceptions with code provided in Supplemental Information 1.

We ran 2,800 simulations for each combination of parameter values. The significance of a given random effect was assessed using likelihood ratio tests (LRTs) between models with and without the focal random effect. To correct for the known conservatism of the LRT when testing for $\sigma^2 = 0$ (due to a null value on the boundary of parameter space), we adopted the standard correction of dividing all $p$-values by 2 (*Pinheiro & Bates, 2001*; *Verbeke & Molenberghs, 2000*; *Fitzmaurice, Laird & Ware, 2004*; *Zuur et al., 2009*). This correction was appropriate for all $p$-values because each LRT compared models that differed in only a single degree of freedom. Power is estimated as the percentage of simulations that provide a corrected $p$-value smaller than 0.05. We insured the validity of a nominal $p$-value of 0.05 by confirming that 2,800 simulations of a scenario with equivalent standard deviations in both treatments did not result in rejecting the null hypothesis more than 5% of the time. Under extremely low numbers of individuals (∼2–4) power to detect differences in the null case exceeded 5% (∼10–15%), possibly inflating power in these cases. Regardless, random effects cannot be reliably estimated with such low sample sizes and therefore in most cases such experimental designs should be avoided.

## Scenario 1: Determining the optimal sampling scheme

Most researchers face limitations imposed by time, money and access to samples, and are therefore confronted with the question of how resources should be divided between

individuals and measures within individuals. To investigate the optimal allocation of sampling effort between the number of individuals and number of observations per individual, we simulated two data sets for each variance comparison (see Table 1 for a summary of all simulations).

First, using three hypothetical total numbers of Bernoulli observations *per treatment* (total sample size per treatment, $TSS_T$), we manipulated either the ratio of individuals to sampling occasions ($\sigma^2_{0k}$ and $\sigma^2_{1k}$), or the ratio of individuals to Bernoulli observations within sampling occasions ($\sigma^2_{vk}$). For comparisons of $\sigma^2_{0k}$ and $\sigma^2_{1k}$ we manipulated the ratio of individuals to sampling occasions, holding the number of Bernoulli observations constant at 5, because power follows a non-monotonic pattern across these ratios for $\sigma^2_{0k}$ and $\sigma^2_{1k}$ (Figs. 2 and 3). Conversely, for comparisons of $\sigma^2_{vk}$ we manipulated the ratio of individuals to Bernoulli observations and held the number of sampling occasions constant at 5 because power follows a non-monotonic pattern across ratios of individuals to Bernoulli observations for $\sigma^2_{vk}$ (Fig. 4). For comparisons of $\sigma^2_{0k}$, and $\sigma^2_{vk}$ we simulated $TSS_T$ of 600, 1,200 and 2,400, and for comparisons of $\sigma^2_{1k}$ $TSS_T$ were 300, 600, and 1,200. For example, for $b_{1ik}$ with a $TSS_T$ of 300, the most extreme ratios were 30 individuals with 2 sampling occasions and 2 individuals with 30 sampling occasions. While using only 2 samples for a grouping variable (individuals) is never suggested for a random effect, we include this combination as an illustration of the low power that results from an ill-conceived sampling scheme. For each variance comparison we simulated three different effect sizes (2, 2.5, and 3 fold difference in standard deviation by treatment).

Next, we simulated data sets with increasing numbers of Bernoulli observations for comparisons of $\sigma^2_{0k}$ and $\sigma^2_{1k}$ (Figs. 5A and 5B) and with increasing numbers of sampling occasions for comparisons of $\sigma^2_{vk}$ (Fig. 5C). For these simulations we used 1, 3, 5, 10 and 15 Bernoulli observations or sampling occasions. Ratios of individuals to sampling occasions ($\sigma^2_{0k}$ and $\sigma^2_{1k}$) or individuals to Bernoulli observations ($\sigma^2_{vk}$) followed the intermediate $TSS_T$ from the simulations described above. For example, for comparisons of $\sigma^2_{0k}$ we simulated 1, 3, 5, 10 and 15 Bernoulli observations for ratios of individuals to sampling occasions ranging from 120:2 to 2:120. For all comparisons we simulated data using an effect size of a 2.5 fold difference in standard deviation by treatment.

In all Scenario 1 simulations, both $\beta_0$ and $\beta_1$ were constrained to a single value for all treatments. For comparisons of among-individual variation in intercept no environmental covariate was used causing each sampling occasion to occur in the same conditions. Additionally, $\sigma^2_{vk}$ was held constant among treatments. For comparisons of among-individual variation in slope we held $\sigma^2_{vk}$ constant. Finally, for comparisons of within-individual variation in intercept, no environmental covariate was included and $\sigma^2_{0k}$ was held constant among treatments. All parameter values used in simulations for both Scenarios can be found in Table S1.

Our goal in Scenario 1 was to isolate changes in a single variance parameter, but exploration of the dependence among multiple variance components and the mean may be warranted if it is relevant for a specific problem. Incorporating concurrent changes in intercept, slope and overdispersion parameters can be easily implemented with slight

Kain et al. (2015), *PeerJ*, DOI 10.7717/peerj.1226

**Table 1 Parameter values for all simulations.** For example, Scenario 1: Fig. 2C illustrates power to detect differences in $\sigma_{0k}^2$ across ratios of individuals to sampling occasions with a $\mathrm{TSS}_T$ of 2,400 at effect sizes of 2×, 2.5×, and 3×difference in standard deviation by treatment.

| Target variance | $\sigma_{0k}^2$ | | | | | $\sigma_{1k}^2$ | | | | | $\sigma_{vk}^2$ | | | | |
|---|---|---|---|---|---|---|---|---|---|---|---|---|---|---|---|
| Scenario | 1 | | | | 2 | 1 | | | | 2 | 1 | | | | 2 |
| Figure | 2A | 2B | 2C | 5A | 6A | 3A | 3B | 3C | 5B | 6B | 4A | 4B | 4C | 5C | 6C |
| Parameter | Sampling occasions | | | Bernoulli obs | $\sigma_{vk}^2$ | Sampling occasions | | | Bernoulli obs | $\sigma_{vk}^2$ | Bernoulli observations | | | Sampling occasions | $\sigma_{0k}^2$ |
| $\mathrm{TSS}_T$ | 600 | 1,200 | 2,400 | 240–3,600 | 2,400 | 300 | 600 | 1,200 | 120–1,800 | 1,200 | 600 | 1,200 | 2,400 | 240–3,600 | 2,400 |
| # Individuals | 2–60 | 2–120 | 2–240 | 120–2 | 2–240 | 2–30 | 2–60 | 2–120 | 60–2 | 2–120 | 2–60 | 2–120 | 2–240 | 2–120 | 2–240 |
| # Sampling occasions | 60–2 | 120–2 | 240–2 | 2–120 | 240–2 | 30–2 | 60–2 | 120–2 | 2–60 | 120–2 | 5 | 5 | 5 | 1–15 | 5 |
| # Bernoulli observations | 5 | 5 | 5 | 1–15 | 5 | 5 | 5 | 5 | 1–15 | 5 | 60–2 | 120–2 | 240–2 | 120–2 | 240–2 |
| Effect sizes | 2; 2.5; 3 | 2; 2.5; 3 | 2; 2.5; 3 | 2.5 | 2.5 | 2; 2.5; 3 | 2; 2.5; 3 | 2; 2.5; 3 | 2.5 | 2.5 | 2; 2.5; 3 | 2; 2.5; 3 | 2; 2.5; 3 | 2.5 | 2.5 |

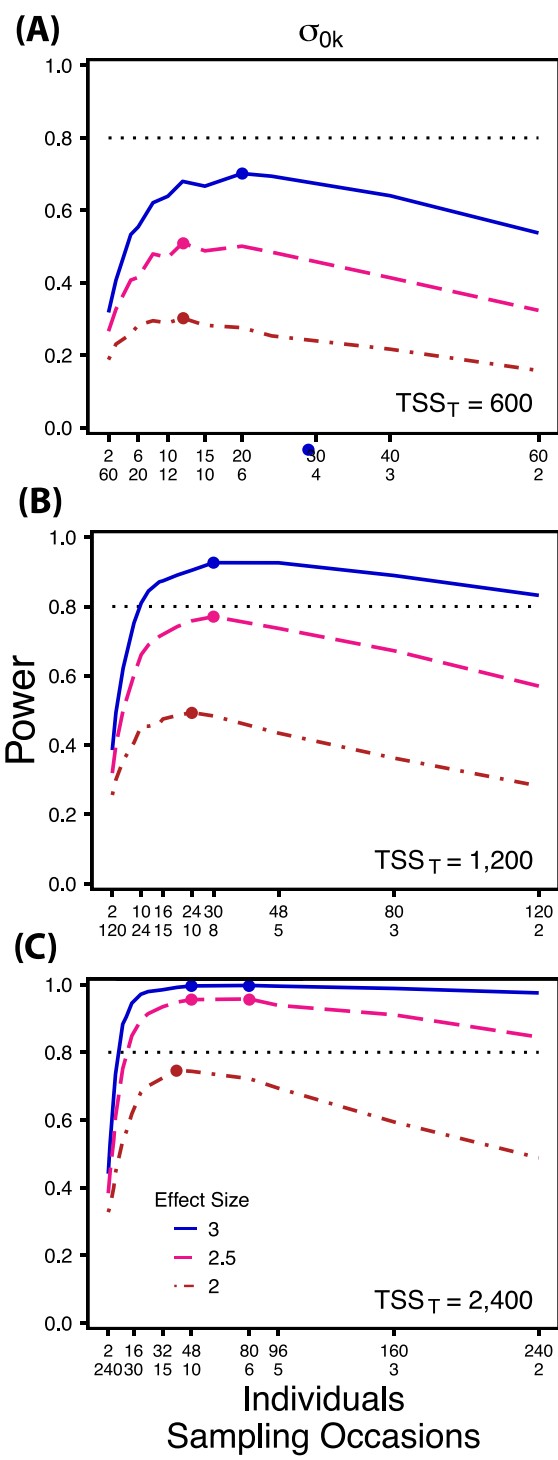

**Figure 2** **Power to detect differences by treatment in among-indiviudal variation in intercept.** Power to detect differences in $\sigma_{0k}$ for three effect sizes (ratio of $\sigma_{0k}$ between treatments) and three $TSS_T$ (total sample size per treatment). Colored circles indicate the ratio of individuals to sampling occasions that optimizes power for each effect size. Each scenario was simulated with 5 Bernoulli observations per sampling occasion.

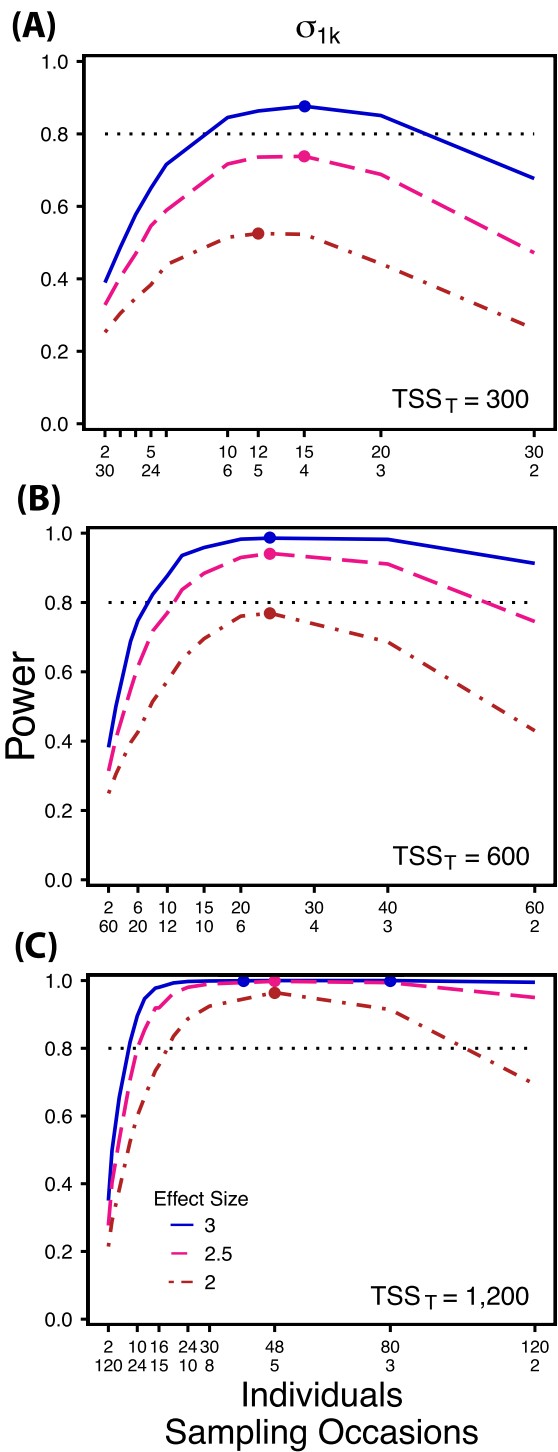

**Figure 3 Power to detect differences by treatment in among-indiviudal variation in slope.** Power to detect differences in $\sigma_{1k}$ for three effect sizes and three $\text{TSS}_T$. Colored circles indicate the ratio of individuals to sampling occasions that optimizes power for each effect size. Each scenario was simulated with 5 Bernoulli observations per sampling occasion.

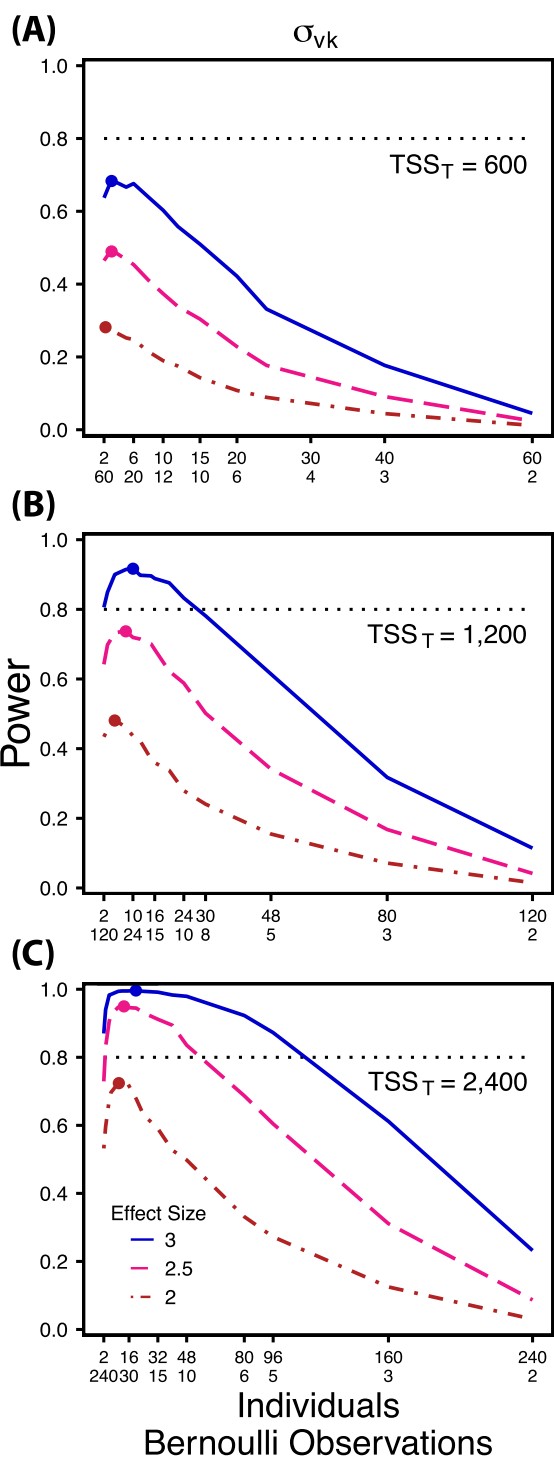

**Figure 4 Power to detect differences by treatment in within-indiviudal variation in intercept.** Power to detect differences in $\sigma_{vk}$ for three effect sizes and three $\text{TSS}_T$. Colored circles indicate the ratio of individuals to Bernoulli observations that optimizes power for each effect size. Each scenario was simulated with 5 sampling occasions.

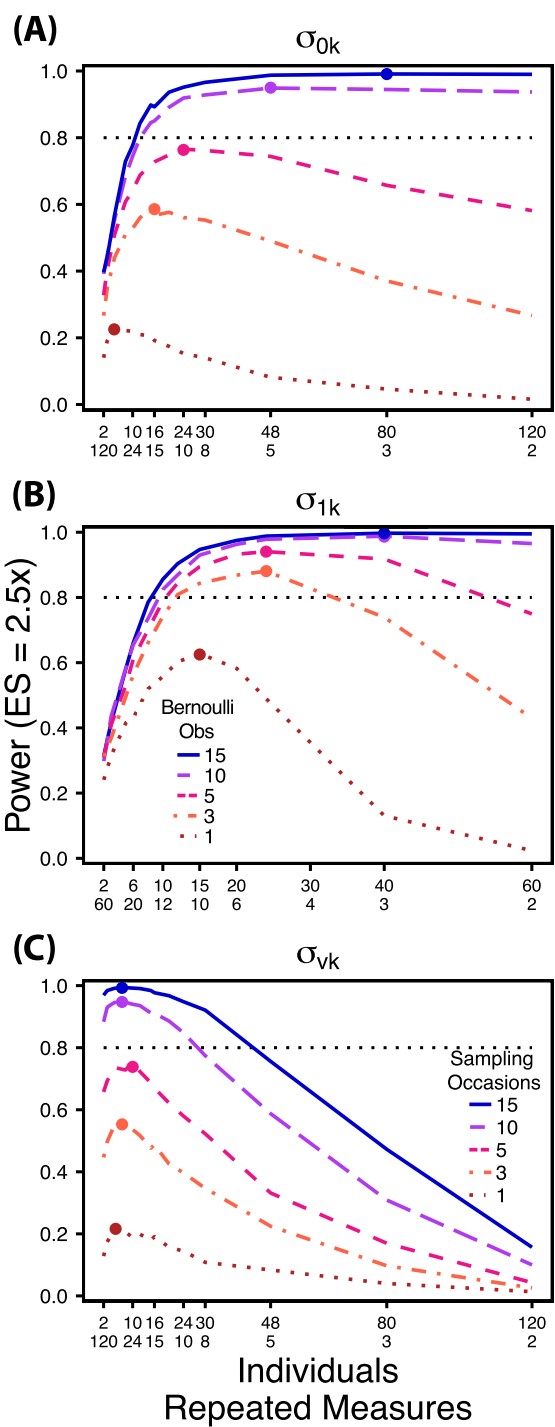

**Figure 5 Power under increasing Bernoulli observations or sampling occasions.** Power to detect differences in $\sigma_{0k}$ (A) and $\sigma_{1k}$ (B) under increasing Bernoulli observations per sampling occasion; $\sigma_{vk}$ (C) under increasing sampling occasions. In (A) and (B) ratios of individuals to sampling occasions follow Figs. 2B and 3B respectively. In (C) ratios of individuals to Bernoulli observations follows Fig. 4B. In (A) and (B) colored circles indicate the ratio of individuals to sampling occasions that optimizes power for each level of Bernoulli observations. In (C) colored circles indicate the ratio of individuals to Bernoulli observations that optimizes power for each level of sampling occasions.

modifications to the code presented in the online supplement. We show initial results of relaxing some of these assumptions in Scenario 2, but full exploration of these possibilities are beyond the scope of this paper.

### Scenario 2: Measuring the ratio of overdispersion to effect size

Decreasing the ratio of the variance in the target random effect to total variance influences power to detect differences in the target variance among treatments. Therefore, we simulated four levels of "noise" (magnitude of non-target random effect variance) assuming a Normal distribution with increasing standard deviations (0.1, 0.5, 1.0, 2.0) (Fig. 6). These correspond to ratios of target variance parameter effect size to non-target variance of 25:1, 5:1, 5:2, and 5:4. For comparisons of $\sigma_{0k}^2$ and $\sigma_{1k}^2$, "noise" was simulated with increasing variation in within-individual variation ($\sigma_{vk}^2$), while for comparisons of $\sigma_{vk}^2$ noise was simulated with among-individual variation in intercept ($\sigma_{0k}^2$). For each variance parameter ratios of individuals to repeated measures followed the largest $\mathrm{TSS}_T$ sampling scheme used in Scenario 1 and an ES of a 2.5× difference in standard deviation by treatment.

## RESULTS

### Scenario 1: Determining the optimal sampling scheme

Power to detect differences between treatments for each variance component increases with total sample size ($\mathrm{TSS}_T$) and effect size (ES) (Figs. 2–5). For a given $\mathrm{TSS}_T$ power depends on the ratio of the number of individuals to the number of repeated measures per individual; however, the optimal ratio of individuals to repeated measures varies depending on $\mathrm{TSS}_T$ and target variance parameter. For example, power to detect both $\sigma_{0k}^2$ and $\sigma_{1k}^2$ is non-monotonic across ratios of individuals to sampling occasions (Figs. 2 and 3), but is an increasing function of the number of Bernoulli observations within sampling occasions (Figs. 5A and 5B). Additionally, for each variance parameter the ratio of individuals to repeated measures within individuals that maximizes power is dependent on both $\mathrm{TSS}_T$ and ES. As $\mathrm{TSS}_T$ and ES increases, greater numbers of individuals relative to repeated measures within individuals leads to higher power for each variance parameter.

At a low sample size ($\mathrm{TSS}_T = 600$) (Fig. 2A) power to detect $\sigma_{0k}^2$ is maximized at a ratio of individuals to sampling occasions of 6:5 at smaller effect sizes (2×, 2.5×) and 10:3 at a large effect size (3×). Under a larger sample size and a small effect size ($\mathrm{TSS}_T = 2,400$, ES = 2×) (Fig. 2C) power is maximized at a ratio of approximately 2:1, while under a large sample size and large effect size ($\mathrm{TSS}_T = 2,400$, ES = 2.5×, 3×) (Fig. 2C), power is maximized at ratios ranging from approximately 5:1 to 13:1.

At a low sample size and effect size ($\mathrm{TSS}_T = 300$, ES = 2×), power to detect $\sigma_{1k}^2$ is maximized at a ratio of 12:5 (Fig. 3A), while larger sample sizes and effect sizes (e.g., $\mathrm{TSS}_T = 600$, ES = 2.5×, 3.0×; $\mathrm{TSS}_T = 1,200$, ES = 2.5×, 3×) favor ratios heavily weighted towards having more individuals versus more repeated measures (ratios ranging from approximately 5:1 to 10:1; Figs. 3B and 3C). Power to detect $\sigma_{1k}^2$ is higher overall and less sensitive to deviations from the optimum ratio than power to detect $\sigma_{0k}^2$ (Fig. 3).

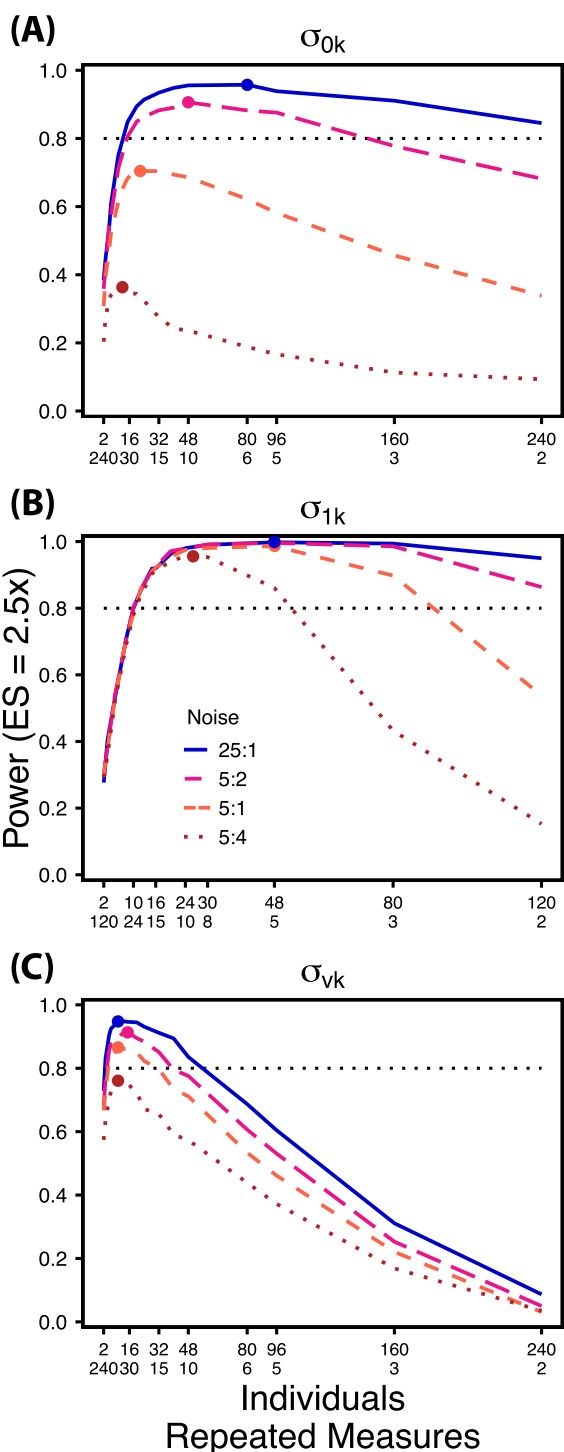

**Figure 6 Power under increasing non-target variation.** Power to detect differences in $\sigma_{0k}$ (A) and $\sigma_{1k}$ (B) under increasing variation in $\sigma_{vk}$; $\sigma_{vk}$ (C) under increasing variation in $\sigma_{0k}$. Noise is given as the ratio of effect size to variation in the non-target variance parameter. In (A) and (B) ratios of individuals to sampling occasions follow Figs. 2C and 3C respectively. In (C) ratios of individuals to Bernoulli observations follows Fig. 4C. Colored circles indicate the ratio of individuals to sampling occasions (A, B) or Bernoulli observations (C) that optimizes power for each level of noise.

Power to detect $\sigma_{vk}^2$ follows a strikingly different pattern than $\sigma_{0k}^2$ and $\sigma_{1k}^2$. Power to detect $\sigma_{vk}^2$ is non-monotonic across ratios of individuals to the number of Bernoulli observations within sampling occasions (Fig. 4), and is an increasing function of the number of sampling occasions (Fig. 5C). At low sample sizes (e.g., $\text{TSS}_T = 600$) power to detect $\sigma_{vk}^2$ is maximized by devoting nearly all of the available resources to repeated measures within individuals (ratios of approximately 1:30 to 3:40; Fig. 4A); however, at a large sample size and effect size (e.g., $\text{TSS}_T = 2{,}400$, ES = 3.0) power is maximized at a ratio of individuals to Bernoulli observations of approximately 5:6 (Fig. 4C).

### Scenario 2: Power under increasing non-target random effect variance

Power to detect differences in variance components is strongly affected by the proportion of total variance that can be attributed to the target variance component (Fig. 6). Increasing variance in non-target random effects decreases power to detect differences in the target variance parameter by treatment. However, the ratio of target to non-target variance does not alter the optimal ratio of individuals to repeated measures for the target variance comparison (Fig. 6). Figure 6A demonstrates that power to detect $\sigma_{0k}^2$ decreases substantially as the magnitude of within-individual variation increases. Detecting differences in $\sigma_{1k}^2$ depends only on total random effect variation at extreme ratios of individuals to sampling occasions (e.g., 80:3) (Fig. 6B). Finally, detection of $\sigma_{vk}^2$ is largely independent of the magnitude of among-individual variation at large ratios of ES to non-target variance, as indicated by overlapping curves in Fig. 6C. However, when among-individual variation in intercept is very large (Fig. 6C: Red curve), power to detect $\sigma_{vk}^2$ decreases because individual mean responses approach 0 or 1, reducing the amount of detectable within-individual variation.

## DISCUSSION

The power analyses presented here establish a framework for designing experiments focused on detecting differences in variance components by treatment using GLMMs. These results should serve as a baseline upon which researchers can expand to address their own specific problems. Nevertheless, our findings reveal some important general trends that should be considered when designing experiments. Our results demonstrate heterogeneity in power across sampling schemes (ratio of individuals to repeated measures and partitioning of repeated measures into sampling occasions and Bernoulli observations), and differences in which sampling scheme maximizes power for different components of variance (Figs. 2–5). As expected, power declines rapidly for low sample sizes and small effect sizes (Figs. 2–4). However, for large $\text{TSS}_T$ and relatively large effect sizes (3 SD difference between treatments), >80% power is retained across many different combinations of individuals to repeated measures for each component of variance (Figs. 2–5). Not surprisingly, power to detect differences in the target random effect declines with increasing variance in the non-target random effects (Fig. 6).

Power to detect $\sigma_{0k}^2$ is non-monotonic across ratios of individuals to sampling occasions, and is an increasing function of the number of Bernoulli observations per sampling

occasion. Power is maximized with ratios weighted towards having more individuals (Fig. 2), and quickly declines with alternative sampling ratios when total sample sizes and effect sizes are small. The analyses are however more robust to deviations from this ratio when $\text{TSS}_T$ and ES are large (Fig. 2C). Finally, of all the random effect parameters we analyzed, power to detect $\sigma_{0k}^2$ is the most sensitive to the amount of "noise" present in the model, decreasing rapidly with increasing within-individual variation (Fig. 6).

Power to detect $\sigma_{1k}^2$ is also non-monotonic across ratios of individuals to sampling occasions, and is maximized with a ratio of individuals to sampling occasions ranging from 2:1 to 5:1 as $\text{TSS}_T$ increases (Fig. 3). On average, testing for differences in $\sigma_{1k}^2$ are more powerful than for $\sigma_{0k}^2$ across all sampling schemes and ES (Figs. 2 and 3), and requires fewer samples to obtain 80% power.

Finally, power to detect $\sigma_{vk}^2$ is non-monotonic across ratios of individuals to Bernoulli observations and is an increasing function of the number of sampling occasions. Depending on sample size, sampling schemes ranging from maximizing Bernoulli observations to ratios of individuals to Bernoulli observations of 1:2 maximizes power (Fig. 4). Unlike $\sigma_{0k}^2$, power to detect $\sigma_{vk}^2$ is largely independent of additional variance in the model (Fig. 6C), such that power to detect $\sigma_{vk}^2$ is nearly equivalent at all levels of $\sigma_{0k}^2$ except under the case of extreme values of $\sigma_{0k}^2$.

Collectively these results indicate the importance of clearly defining a biological question, designating the focal random effect, and knowing the expected magnitude of total variation when determining the appropriate experimental sampling design and $\text{TSS}_T$. Even at larger effect sizes, failure to account for system noise can lead to insufficient power and a failed experiment. Our findings should serve as a strong warning to empiricists interested in variance components that power analyses should be performed when designing experiments in order to overcome the problems of overall low power, large heterogeneity in power to detect different variance components, and heterogeneity in sampling scheme required to optimize power.

By introducing new strategies for analyzing variance among treatments we hope to inspire novel experimental designs in ecology and evolution. For example, the power analyses presented here can inform the design of experiments aimed at quantifying heterogeneous within-individual variation by environment, which may lead to novel insights on the adaptive significance of within-individual variation (*Westneat, Wright & Dingemanse, 2014*).

In addition, these analyses answer the calls of researchers over the last decade for methods to investigate effects of treatment level variance on the variance of dependent variables (*Benedetti-Cecchi, 2003*). Transitions from one discrete environment to another (e.g., presence or absence of predators) are often classified as a form environmental variation, but switching between two distinct but relatively constant environments does not reflect environmental variation *per se*, such as temporal changes in the magnitude, pattern, and/or frequency of the environmental over time (*Benedetti-Cecchi, 2003*; *Benedetti-Cecchi et al., 2006*; *Miner & Vonesh, 2004*; *Lawson et al., 2015*). When this form of environmental variation is manipulated or natural variation exploited in an experimental

context, within-individual variation can be described as the variable response of individuals to this variation in the environment. In this context, within-individual variation may itself be a form of phenotypic plasticity, and may have profound implications for understanding the evolution of environmentally induced plasticity, and the evolution of labile traits generally (*Stamps, Briffa & Biro, 2012*; *Biro & Adriaenssens, 2013*; *Westneat, Wright & Dingemanse, 2014*).

## Further considerations

### Heterogeneous within-individual variation

In our power analyses we have made a few important simplifying assumptions. First, we assume that within-individual variation in both intercept and slope is homogenous among individuals within the same treatment. Additionally, we assume homogeneity of within-individual variance across an environmental gradient. However, these assumptions may not be true for some natural or experimental populations. In fact, it has recently been proposed that assessing the magnitude of variation in within-individual error variance within a single individual across an environmental gradient or among individuals exposed to the same environment/treatment is an important metric that may help to explain the evolution of plasticity (*Cleasby, Nakagawa & Schielzeth, 2015*; *Westneat, Wright & Dingemanse, 2014*). Power to detect differences in the magnitude of among-individual variation in within-individual variation by treatment (*Cleasby, Nakagawa & Schielzeth, 2015*) and heterogeneity of variance across an environmental gradient are interesting research questions that deserve attention, but are beyond the scope of this article. We also note that practicality limits exploration of increasingly complicated scenarios, despite their conceivable statistical feasibility and intrinsic charm due to complex novelty.

### Covariance among intercept, slope, and variance components

All of our simulations assessed power to detect differences in a single target variance comparison between treatments, holding all other variance parameters constant (Table S1). However, manipulating non-target variation generates additional variation that is expected to decrease power to detect differences in the target variance parameter. Because we assumed no slope variation in models where intercepts were allowed to vary and no intercept variation in the models focused on variation in slopes, we did not discuss power to detect covariance terms. However, these parameters can co-vary and the covariation among these parameters may contain a wealth of biologically relevant information. For example, covariation between phenotypic plasticity and within-individual variation may be tightly linked via developmental tradeoffs, which can lead to greater developmental instability in highly plastic individuals (*Tonsor, Elnaccash & Scheiner, 2013*). Indeed, it is not known whether an individual's reaction norm slope and within-individual variation around that reaction norm are always linked or if these relationships can be context-dependent. Similarly, we do not know if stronger behavioral responses lead to greater canalization of behavior. Understanding how to parameterize GLMM and how to optimize experiments to detect these covariances will be a useful step toward advancing evolutionary theory on adaptive, maladaptive and random patterns of variation.

Covariance between intercept and slope has been described extensively in theoretical papers and has been explored in earlier power analyses for LMM (*Dingemanse & Dochtermann, 2013*); however, empirical studies documenting significant covariance between these parameters remain rare (*Mathot et al., 2011*; *Dingemanse et al., 2012*). While covariance among these parameters may be uncommon, it is also likely that most experiments have insufficient power to detect such covariance. Additional analyses that determine power to detect significant differences in intercept and slope covariation for GLMMs is another important step considering the lack of current evidence for covariation reported in the literature.

### Within-individual variation in slope

Research, including ours, on among-individual variation in plasticity assumes fully repeatable plasticity within each individual, causing among-individual differences in phenotypic plasticity to be calculated using a single reaction norm for each individual (*Dingemanse & Wolf, 2013*). However, quantifying only a single reaction norm for each individual fails to capture any potential variation in plastic responses within an individual around its mean reaction norm, which may inflate estimates of among-individual variation and mask important variation that is subject to selection (*Dingemanse & Wolf, 2013*). Despite the reasonable assumption that each experimental individual would exhibit variation in their reaction norm if it were repeatedly measured, we are aware of no studies that demonstrate repeatable behavioral plasticity for a single individual when assessed multiple times.

### Heterogeneity in sampling scheme and environment

In our simulations all individuals were measured an equal number of times and all treatments contained the same number of individuals, a luxury often not available to empiricists that often deal with missing data and unbalanced designs. Intuitively, unbalanced sampling schemes will lower the power to detect among-individual variation (*Van de Pol, 2012*); however we do not know the rate at which statistical power is lost with the magnitude of imbalance for a particular sampling design. In highly unbalanced designs or when data have many missing observations state-space models may be a more powerful alternative to GLMMs for separating different types of variability (*Schnute, 1994*). Future research should follow the lead of Van de Pol (2012) to determine how power to assess differences in variance for GLMM is affected by incomplete sampling, specifically when only a single measure is available for some individuals.

### Experiments with more than two treatments

Finally, these power analyses were created for a two-treatment scenario—"homogenous" environmental variation treatment and a "variable" environmental variation treatment. However, it is commonplace to have more than two treatments. Fortunately, our framework for conducting power analyses can be easily generalized for exploring power for experiments with more than two treatments (see Supplemental Information 1). In addition, syntax for the `lme4` package in R for specifying GLMM is highly flexible and can be written to restrict variance components to be the same in any number of treatments, while

unique variance estimates can be obtained for any other given treatment. For example in a four treatment experiment composed of four levels of predator cue, two variance estimates could be obtained for among-individual variation (e.g., a single estimate for the three treatments with the lowest levels of predator cue and one estimate for the highest level of predator cue). As in the two-treatment scenario, differences in variance among treatments in a multi-treatment scenario can be evaluated with a likelihood ratio test.

## CONCLUSIONS

Despite the ubiquity of random intercepts and slopes GLMMs in ecology, evolution, and behavior, the use of GLMMs to compare variance components among populations or among experimental treatments is rare. We hope the power analyses presented here will spur novel empirical research and assist readers in constructing appropriate experimental designs and statistical models to test how variance components are shaped by ecological and evolutionary processes. We emphasize a clearly defined biological question and designation of the focal random effect when designing experiments for this application due to unique ratios of individuals to repeated measures required to optimize power for each variance parameter and low overall power (in most cases >1,000 total Bernoulli observations per treatment needed to achieve 80% power). Finally, we call for future work analyzing the accuracy and precision of estimates comparing random effects by treatment for GLMMs (which our code facilitates) similar to the work of *Moineddin, Matheson & Glazier (2007)* and *Van de Pol (2012)* on the accuracy and precision of random effects estimates. As Van de Pol points out, just because power is high does not ensure the accuracy and precision of estimates.

## ACKNOWLEDGEMENTS

We thank the lab members of Dr. Michael McCoy's lab and Dr. Krista McCoy's lab for helpful comments during the preparation of this manuscript. We also thank Daniel Hocking and one anonymous reviewer for many helpful comments and suggestions that improved the clarity of this manuscript.

### Funding

This research was supported through a new faculty start up grant from East Carolina University. The funders had no role in study design, data collection and analysis, decision to publish, or preparation of the manuscript.

### Grant Disclosures

The following grant information was disclosed by the authors:
East Carolina University.

### Competing Interests

The authors declare there are no competing interests.

## Author Contributions

- Morgan P. Kain conceived and designed the experiments, performed the experiments, analyzed the data, contributed reagents/materials/analysis tools, wrote the paper, prepared figures and/or tables, reviewed drafts of the paper.
- Ben M. Bolker and Michael W. McCoy conceived and designed the experiments, contributed reagents/materials/analysis tools, reviewed drafts of the paper.

## Supplemental Information

Supplemental information for this article can be found online at http://dx.doi.org/10.7717/peerj.1226#supplemental-information.

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
