# Peer review of "A practical guide and power analysis for GLMMs: detecting among treatment variation in random effects"

_PeerJ, doi:10.7717/peerj.1226_

## Round 0.1 · original submission · Minor Revisions

I have received two very positive reviews and have read the paper myself. We all agree that this is a very nice paper that requires only minor revisions. The paper is exceptionally clear and should be very valuable for increasing understanding and utility of GLMMs. Please consider the comments of the reviewers.

I agree that there is no reason to limit the application of the models to behavioral data.

In the missing observation part of the discussion it would be useful to mention state space models.

I found the model description around lines 192 clearer than the first description around line 109.

My new favorite quote is 'intrinsic charm due to complex novelty'.

Reviewer 1 ·

Basic reporting

1. Authors “hope to inspire novel experimental designs in ecology and evolution” (last sentence of abstract), but use “vocabulary and examples from behavioral ecology” (line 70). Given that this is a simulation-based study and binomial data are common in other branches of ecology and evolution, it might help include other examples outside of behavioral ecology. The simulation result is clearly applicable to habitat occupancy, animal movement and foraging/feeding success, etc. that typically produce binomial data. I don’t think that it is absolutely necessary to go beyond behavioral ecology but it would make the paper appealing to a wider range of audience.
2. Precise and consistent definition. GLMMs are complex and thus definition and terminology should be explained precisely and consistently. I noticed several descriptions that led to unnecessary confusions:
- Authors describe that the first set of simulations tested “the ratio of groups to total observations within groups (n*j), and consider different partitions of n and j.” (lines 60-61). But it is clear on Figures 2 & 3 that the number of Bernoulli observations (n) was fixed at 5, while changing the combinations of individuals and sampling occasions. This description needs to be changed.
- “j” was used to index sampling occasions on line 82, but then it was again used to index environment on line 108. Are sampling occasions and environment synonymous? I guess it may be true when covariates are not included? At any rate, I would use one word consistently throughout the manuscript.
- The word “treatment” is frequently used in the manuscript and I now understand that it is referred to a group of individuals subject to the same condition. Perhaps this is a common word in behavioral ecology, but it took a couple of reading to confirm what it meant (I am not a behavioral ecologist). This is related to the first comment above, but it might be helpful to define terminology at the first mention, especially if authors decide to continue to use vocabulary and examples from behavioral ecology but want to make this paper relevant to a wider range of ecologists.
- On line 85, “variation among sampling occasions in the probability an individual displays a behavior” – “among” should be “within”, correct?

Experimental design

No Comments. The range of simulations conducted seems reasonable.

Validity of the findings

This comment is not quite related to validity of the findings, but figures showing simulation results (Figs. 2-6) are sometimes difficult to interpret because the peak is not easy to distinguish on several curves. The text in Results section should help, but not all always. For example, on lines 317-320, “Power to detect σ2 0k is maximized at a ratio of individuals to repeated measure of approximately 6:5 under low sample size (TSSt = 600) (Figure 2A)” – this appears to depend on effect size. On Figure 2A, the blue line appears to peak at 6:20, but pink and red lines appear to peak at 12:10 ratios. As another example, on Figure 3C, authors report that “larger sample sizes (TSSt=600, 1200) favor a ratio heavily weighted towards having more individuals (approximately 5:1) versus more repeated measures.” This is difficult to tell because the flat portion of the blue and pink curves are so long. Overall, I did not find Results section straight-forward to follow. Perhaps, identifying the peak of curves using dot or point on each panel would help greatly.

Additional comments

I enjoyed reading this manuscript. I agree that a focus on variance by treatment is important and timely. Discussion on further consideration of potential practical scenarios is helpful. A major take home message in my mind is that the best sampling scheme can differ by target variance parameter. This helps and forces the researcher to think about what the most important variance to estimate is before a specific study commences and should lead to a more informed decision on sampling designs.

·

Basic reporting

No Comments

Experimental design

No Comments

Validity of the findings

No Comments

Additional comments

Also attached as a PDF

## Overall Comments

This manuscript presents information on two aspects of generalized linear mixed models not well appreciated in the ecological, evolutionary, or behavioral literature. The first is a way to compare within- and among-group variation using GLMMs. This is particularly useful in behavioral and evolutionary studies; situations they describe well. The second part of the paper is a description and example of how to conduct a power analysis for designing complex experiments with repeated samples of individuals within groups or treatments. This is a very common experimental design. I have talked with other colleagues about this issue in the past without much consensus on how to best allocate effort/subjects/treatments/groups. I've previously conducted clunky simulations to assess power, but this paper would have been very useful.

This paper provides valuable information regarding the optimal balance of repeated sampling of individuals and the total number of individuals. While the results are not fully generalizable because of the other factors that influence ability to detect change (effect sizes and variances), the methods are generalizable and sufficient information is provided such that a reader could easily perform their own power analyses.

Many of the findings are intuitive and not surprising. They may even be trite for statisticians. However, for many practicing EEB scientists the manuscript provides some useful general recommendations, but most importantly a framework for systematically considering and simulating variation within and among individuals across treatments. This is valuable in many fields and for a variety of applications. The critical power detection and relative effect sizes and variances, along with sampling limitations, will always necessitate problem-specific power analyses. However this manuscript and associated supplements provide a readable, instructional way forward for graduate students and typical researchers. It also provides (slightly disheartening) information regarding the large amount of sampling and subjects required to detect difference in variance components among groups or treatments.

There are some assumptions required for these analyses. They are all reasonable for some systems and, most importantly, they are well described and acknowledged. Individual readers can decide whether their system sufficient meets these assumptions. In the *Further Considerations* section the authors describe additional complications and potential applications. Although these were beyond the scope of this manuscript, it seems reasonable that a reader could adapt their power analyses to address things like covariance between various terms. This would provide an interesting program for an entrepreneuring graduate student to explore further.

## Introduction

The introduction is exceptionally well written. It lays out need for the paper (problem) and relevant literature in organized, easy-to-understand language. this can be a challenge for technical (mathematically and computationally) papers but the authors do an admirable job. This is critical in a manuscript intended for a general EEB audience.

The only place where more elaborate might be useful is on lines 57-61. Describing in real-world terms what the Bernoulli observations are in contrast to sampling occasions and individuals could be helpful for some readers ("for example, number of times a behavior was displayed during observation period *j* for individual *i*"). This becomes more clear in the final paragraph of the introduction but is worth including earlier as 1 sentence or even parenthetically.

## Methods

I like the layout and description of equations 1-4. My only comment is to consider adding the term "fixed effects" to $\beta_{0k} + \beta_{1k}X_{ij}$ to correspond to the output the *lme4* which many of the readers will be familiar with and which is used in this paper.

L127-128: I am not sure if all readers will understand what is meant by the "variation in individuals' average behavior in the mean-centered environment". It might be worth rewording or providing a more colloquial description as a next sentence. I do really appreciate the real-world example that ends the paragraph.

It is unclear whether the linear mixed model section is needed or if the information could easily be merged with the binomial GLMM section. The GLMM model is not much different, so I'm not sure if it's really valuable having it as 2 sections and introducing LMM separately from GLMM. However, the manuscript flows very nicely as is, so I wouldn't advocate for merging them unless there was another reason to do so.

I appreciate the well laid out description of handling overdispersion L195-201). This has been an issue with mixed models and there has been relatively little in the ecological literature describing the logic behind it (but see Harrison 2014 PeerJ). It is also great to see this term used informatively rather than just as a nuisance parameter when the binomial (or similarly Poisson) distribution just doesn't fit.

L223-224: It would be useful to further describe how these values are "biologically relevant". Maybe some indication of studies with similar probabilities and variances.

## Discussion

It would be worth adding how one would test for differences among more than 2 treatments (as opposed to the LRT) in the final paragraph ~L500.

## Conclusions

This was fine but was somewhat redundant, although not completely, and didn't really feel like conclusions. The note regarding power not ensuring accuracy or precision was good (and first time it was mentioned).

## Figures

**Figure 1:** It would be beneficial to include the terms from the equations relevant to plots A, B, and C (i.e. $\sigma^{2}_{0k}$, $\sigma^{2}_{1k}$, $\sigma^{2}_{\epsilon k}$).

## Additional Comments

I am not aware of peerj's policy regarding the number of in-text citations together. Some journals suggest or require e.g. then a max of 3 citations. In this manuscript there are sometimes 7+ citations at once. As a non-print journal I hope this is acceptable because in a "how to" article like this it is valuable for many readers to have many citations to refer to as they learn and try to integrate this new information with their existing knowledge.

Given the ease of markdown, Rmarkdown (RStudio), and basic text editors, it seems unnecessary to have pasted code with `>` and `+` symbols. It prevents code from easily being pasted back into R scripts. That being said, Supplement 1 is a helpful and necessary part of the paper.

L239: Awkward wording with multiple "did not" statements in succession. Consider rephrasing.

The need for a "recent version" of `lme4` is mentioned in the main text and in the supplement. This is vague. A version number should be indicated (e.g. >1.1-7). Even if the authors do not remember the earliest version that the tools were included in and can't find it on git, the version used in the analysis should be indicated in the text as the minimum one to use.

---

## Round 0.2 · accepted · Accept

Thank you for carefully considering the reviewer's comments. You have adequately addressed all issues raised.